

# Production mechanisms of open heavy flavor mesons in high-multiplicity events

**Marat Siddikov⋆ and Iván Schmidt**

Departamento de Física, Universidad Técnica Federico Santa María,
y Centro Científico - Tecnológico de Valparaíso, Casilla 110-V, Valparaíso, Chile

⋆ Marat.Siddikov@usm.cl

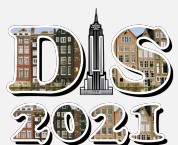

*Proceedings for the XXVIII International Workshop
on Deep-Inelastic Scattering and Related Subjects,
Stony Brook University, New York, USA, 12-16 April 2021*

## Abstract

**In this proceeding we discuss different mechanisms of open-heavy flavor meson production which contribute to inclusive and single diffractive cross-sections. For the case of inclusive production we demonstrate that the three-Pomeron fusion is significant for the $D$-meson production in the kinematics of small transverse momenta $p_T$, as well as in large-multiplicity events. Its inclusion allows to improve significantly the description of experimental data. We also analyzed the diffractive production and found that due to gap survival factors it constitutes 0.5–2 percent of the inclusive production. The expected dependence on event multiplicity in this channel is significantly milder than for inclusive case.**

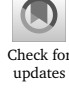

## 1 Introduction

Nowadays the production of heavy mesons play an important role in understanding the dynamics of strong interactions in nonperturbative regime, since in the limit of the infinitely heavy quark mass it becomes possible to use certain perturbative techniques. Recently the attention of theoretical and experimental studies focused on the dependence of the cross-section on multiplicity of hadrons co-produced together with a heavy meson [1–5]. Since lack of statistics presents the largest obstacle for measurement of rare high-multiplicity processes, it is expected that analyzed multiplicity studies could benefit from enhanced luminosity in Run 3 at LHC (HL-LHC mode) [6–8].

Due to Local Parton Hadron Duality the multiplicity of light hadrons is proportional to the multiplicity of partons in collision, so the dependence on multiplicity allows us to estimate the density of partons (mostly gluons) with different rapidities during a collision. As was found

recently by STAR and ALICE [1–4], for the production of $J/\psi$ and $D$-mesons the observed multiplicity dependence is faster than could be expected from pomeron-pomeron fusion. Possibly this effect could be explained by sizable contribution of multipomeron mechanisms (e.g. triple pomeron fusion). While theoretical estimates in the heavy quark mass limit suggest that such corrections should be suppressed as higher twist effects, due to increase of gluon density they might give sizable contribution, thus challenging all the evaluations in the $k_T$- and collinear factorization approach. In the kinematics of very small values of $x$ consideration based on partonic picture might be not reliable at all, and it is more appropriate to use the frameworks with built-in saturation, like *e.g.* the Color Glass Condensate approach and closely related color dipole picture [9–15]. In this approach the cross-sections of different processes might be expressed in terms of the color singlet dipole scattering amplitude known from Deep Inelastic Scattering. This framework gives successful description of the cross-sections of various lepton-hadron and hadron-hadron processes. The generalization of the approach to high-multiplicity events is straightforward and was discussed in [16, 20–24]. For this reason in what follows we will use this approach for study of heavy meson production in LHC kinematics.

This proceeding is structured as follows. In the following Section 2 we briefly summarize the main results for the cross-sections of different production mechanisms (inclusive and single-diffractive). In the Section 3 we discuss extension of the framework for description of high multiplicity events, and compare expected multiplicity dependence with available experimental data. Finally, in Section 4 we draw conclusions.

## 2 Production of heavy hadrons

As we mentioned earlier, the color dipole framework allows to express the cross-sections of various processes via forward dipole amplitudes, convoluted with the wave functions of quarkonia or fragmentation functions of the open heavy flavor mesons. The exact structure of the expressions depends on the process under consideration. The cross-section of heavy flavor production might be written as [24–28]

$$\frac{d\sigma_{pp\to\bar{Q}_iQ_i+X}\left(y,\sqrt{s}\right)}{dy\,d^2p_T} = \int d^2k_T x_1 g\left(x_1, \boldsymbol{p}_T - \boldsymbol{k}_T\right) \int_0^1 dz \int \frac{d^2r_1}{4\pi} \int \frac{d^2r_2}{4\pi} \tag{1}$$
$$\times\; e^{i(r_1-r_2)\cdot\boldsymbol{k}_T}\Psi_{\bar{Q}Q}^\dagger\left(r_2, z, p_T\right)\Psi_{\bar{Q}Q}^\dagger\left(r_1, z, p_T\right)N_M\left(x_2(y); \boldsymbol{r}_1, \boldsymbol{r}_2\right) + (y\to -y),$$

where the variables $y$ and $\boldsymbol{p}_T$ are the rapidity and transverse momenta of heavy quark, $x_{1,2} \approx e^{\pm y}\sqrt{(m_M^2 + \langle p_{\perp M}^2\rangle)/s}$, $z$ is the light-cone fraction of the quark in the dipole, and $(\boldsymbol{r}_1, \boldsymbol{r}_2)$ are the transverse sizes of a dipole in the amplitude and its conjugate. The notation $\Psi_{\bar{Q}Q}(r, z)$ stands for the light-cone wave function of the $\bar{Q}Q$ pair of transverse size $r$ and the light-cone fraction of the quark $z$. The expression (1) includes a nonperturbative object $N_M$ which encodes interactions of a dipole with the target. The structure of this object depends on the quantum numbers of the dipole and production mechanism. While in general it is expected that $N_M$ should be a *linear* superposition of forward dipole amplitudes $N(x, \boldsymbol{r}, \boldsymbol{b})$, the nonlinear dynamics might give rise to the so-called *quadrupole* contributions, which manifest themselves as quadratic (bilinear) terms in the relation between inclusive production cross-sections and dipole amplitudes [29]. Following the BFKL terminology, we will refer to the latter contributions as three-pomeron terms. For the case of inclusive and single diffractive production the explicit expressions for $N_M$ might be found in [25–28]. The three-pomeron contributions in general are not positively defined and cannot be interpreted probabilistically. In what follows we will use for their evaluation the CGC parametrization of the singlet color singlet dipole amplitude available from [30].

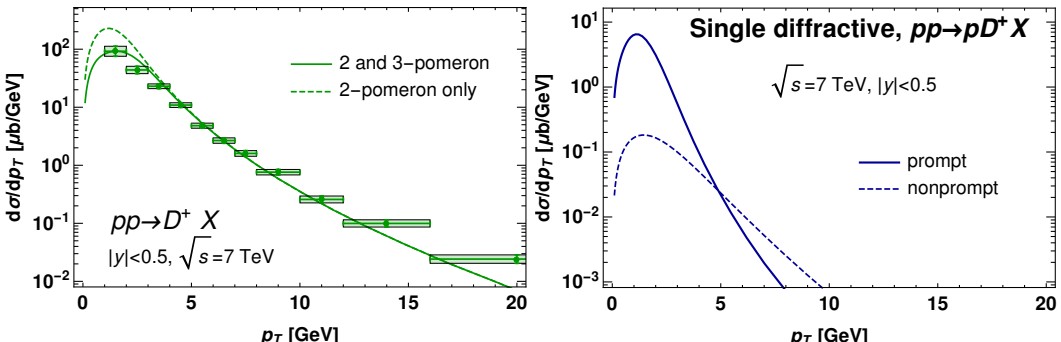

Figure 1: The $D^+$-meson production cross-section as a function of transverse momentum $p_T$. Left and right plots correspond to inclusive and single diffractive production at central rapidities respectively. As explained in the text, the three-pomeron contribution in left-hand side is not positively defined due to interference diagrams, so its inclusion *decreases* the overall result. The experimental data are from [31–33].

In the Figure 1 we have shown our results for the inclusive and single diffractive production in the central kinematics. The inclusion of three-pomeron terms clearly improves the description of the inclusive production cross-section. The single diffractive cross-section is suppressed stronger than inclusive one at large $p_T$ since its dominant contribution is formally a higher twist effect. We found that description of $B$-mesons and non prompt $J/\psi$ meson production in dipole approach has the same level of agreement with data (see [26–28] for details), so we may use this approach for studies of multiplicity.

## 3 Multiplicity dependence

The events with enhanced multiplicity are very rare, for this reason their study requires outstanding luminosity. In the literature the results for multiplicity dependence are conventionally given for the self-normalized double ratio

$$\frac{dN_M/dy}{\langle dN_M/dy \rangle} = \frac{d\sigma_M\left(y, \eta, \sqrt{s}, n\right)/dy}{d\sigma_M\left(y, \eta, \sqrt{s}, \langle n \rangle = 1\right)/dy} \bigg/ \frac{d\sigma_{\mathrm{ch}}\left(\eta, \sqrt{s}, Q^2, n\right)/d\eta}{d\sigma_{\mathrm{ch}}\left(\eta, \sqrt{s}, Q^2, \langle n \rangle = 1\right)/d\eta}, \quad (2)$$

where $n = N_{\mathrm{ch}}/\langle N_{\mathrm{ch}}\rangle$ is the relative enhancement of multiplicity, the variables $y$ and $\eta$ stand for the pseudorapidities of heavy meson and charged particles respectively. The numerator and denominator of (2) correspond to relative multiplicity dependence of the heavy meson ($M$) underproduction and that of the inclusive channel.

As is known from the literature [16, 20–23], in high-multiplicity events the saturation scale $Q_s^2$ increases as

$$Q_s^2\left(x, b; n\right) \approx n\, Q^2\left(x, b\right). \quad (3)$$

For studies of the multiplicity dependence it is important to mention that the modification of the dipole amplitude like (3) should be applied only if the multiplicity enhancement is measured in the rapidity interval between a heavy dipole and the target (otherwise the dominant contribution would come from configurations with $n = 1$). In case of single diffractive production, as well as inclusive production with rapidity-separated $\eta$- and $y$-bins it is clear that only some of the amplitudes will be modified by additional constraints of elevated multiplicity. On the other hand. the widely used setup with overlapping bins in $\eta$ and $y$ would get sizable

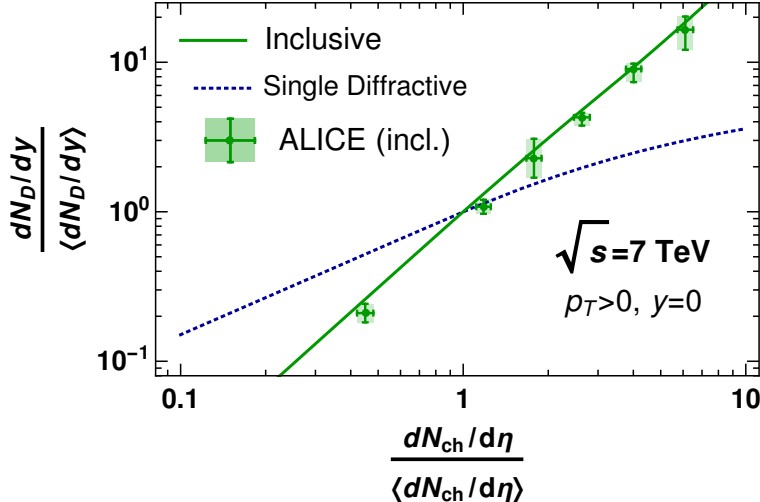

Figure 2: Results for the multiplicity dependence of inclusive and single diffractive production of $D$-mesons in LHC kinematics. For inclusive production we take into account both prompt and nonprompt mechanisms. The experimental data are for the multiplicity dependence of inclusive production from ALICE [1], and it is assumed that all particles are collected at central rapidities in rapidity window $|y|, |\eta| \lesssim 1$.

contribution from the configurations in which the enhanced multiplicity is shared by all the reggeized gluons which participate in the process. Technically this would lead to modification of both gluon density and dipole amplitude in (1). In the Figure 2 we show our expectations for the multiplicity dependence of the inclusive and single diffractive data. We can see that the model provides quite reasonable description for the inclusive production. For the single diffractive production the predicted dependence on multiplicity is milder than for inclusive production due to specifics of this process (in pomeron language, it has only one cut pomeron which contributes).

## 4  Conclusions

In this proceeding we presented the results for the multiplicity dependence of inclusive and single diffractive $D$-meson production. We found that in case of inclusive production inclusion of the three-pomeron contribution improves description of the inclusive data at small-$p_T$, as well as helps to describe the observed data on multiplicity dependence. For single diffractive production we expect milder dependence on multiplicity, since only one pomeron in the cross-section is cut and may contribute to yields of charged particles. This result illustrates that the expected multiplicity dependence is related to the production mechanism of heavy quarks rather than final state interactions which lead to its hadronization (fragmentation) into $D$-mesons. The predictions for the $B$-mesons and nonprompt $J/\psi$ mesons have qualitatively similar dependence on multiplicity and $p_T$ and might be found in [26–28].

## Acknowldgements

We thank our colleagues at UTFSM university for encouraging discussions. This research was partially supported by Proyecto ANID PIA/APOYO AFB180002 (Chile) and Fondecyt (Chile)

grant 1180232.

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
