# Peer review of "Production mechanisms of open heavy flavor mesons in high-multiplicity events"

_SciPost Physics Proceedings, doi:SciPost Phys. Proc. 8, 096 (2022)_

## Round 1 · Referee Report · Anonymous (Referee 1) · 2022-2-10

Report

General observation:
Fig. 1, assuming the solid line is for 2-pomeron and dashed line for 2 and 3-pomeron as described in the figure, I don't see the improvement at small P_T. The dashed curve clearly over predicts the data. This doesn't support the conclusion.

Minor correction:
Abstract: "per cent" -> "percent"
  • validity: -
  • significance: -
  • originality: -
  • clarity: -
  • formatting: -
  • grammar: -

Author:  Marat Siddikov  on 2022-02-11  [id 2191]

(in reply to Report 1 on 2022-02-10)

Dear Editor,

We thank the referee for careful reading of our report. Below we send our replies to the referee's comments and the updated version of our manuscript (attached file "Proceeding_DIS2021_v02.pdf")

Sincerely,
Marat Siddikov

REFEREE:
General observation:
Fig. 1, assuming the solid line is for 2-apomeron and dashed line for 2 and 3-pomeron as described in the figure, I don't see the improvement at small P_T. The dashed curve clearly over predicts the data. This doesn't support the conclusion.

AUTHORS:
In the previous version we had mistakenly set the labels in the left panel of the Figure 1. The dashed line in this plot corresponds to 2-pomeron only contribution, whereas solid line is a sum of 2- and 3-pomeron contributions. The caption of the Figure correctly claims that 3-pomeron contribution is negative, so its inclusion decreases the total result and improves agreement with data (as we can see from the plot, the solid line is closer to the data than the dashed one).

In order to correct this issue, in the resubmitted version we corrected the labels in the left panel of the Figure 1.

REFEREE:
Minor correction:
Abstract: "per cent" -> "percent"

AUTHORS:
We corrected this spelling error in the resubmitted version.

Attachment:

Proceeding_DIS2021_v02.pdf

Author:  Marat Siddikov  on 2022-02-11  [id 2190]

(in reply to Report 1 on 2022-02-10)
Category:
reply to objection
correction

Dear Editor,

We thank the referee for careful reading of our report. Below we send our replies to the referee's comments and the updated version of our manuscript (attached file "Proceeding_DIS2021_Draft_v02.pdf")

Sincerely,
Marat Siddikov

REFEREE:
General observation:
Fig. 1, assuming the solid line is for 2-apomeron and dashed line for 2 and 3-pomeron as described in the figure, I don't see the improvement at small P_T. The dashed curve clearly over predicts the data. This doesn't support the conclusion.

AUTHORS:
In the previous version we had mistakenly set the labels in the left panel of the Figure 1. The dashed line in this plot corresponds to 2-pomeron only contribution, whereas solid line is a sum of 2- and 3-pomeron contributions. The caption of the Figure correctly claims that 3-pomeron contribution is negative, so its inclusion decreases the total result and improves agreement with data (as we can see from the plot, the solid line is closer to the data than the dashed one).

In order to correct this issue, in the resubmitted version we corrected the labels in the left panel of the Figure 1.

REFEREE:
Minor correction:
Abstract: "per cent" -> "percent"

AUTHORS:
We corrected this spelling error in the resubmitted version.

Attachment:

Proceeding_DIS2021_Draft_v02.pdf

Anonymous on 2022-02-11  [id 2192]

(in reply to Marat Siddikov on 2022-02-11 [id 2190])

Thanks for making the revision, and "Yes"! The 3 pameron contribution significantly improve the fitting quality.

Recommend to accept for publication.

---

## Round 2 · Author Response

Please find below the resubmitted version of our proceeding to DIS2021 issue. In this version we addressed all the issues raised in the previous report of the referee.
Sincerely,
Marat Siddikov

---

## Round 2 · List of Changes

REFEREE:
General observation:
Fig. 1, assuming the solid line is for 2-apomeron and dashed line for 2 and 3-pomeron as described in the figure, I don't see the improvement at small P_T. The dashed curve clearly over predicts the data. This doesn't support the conclusion.
AUTHORS:
In the previous version we had mistakenly set the labels in the left panel of the Figure 1. The dashed line in this plot corresponds to 2-pomeron only contribution, whereas solid line is a sum of 2- and 3-pomeron contributions. The caption of the Figure correctly claims that 3-pomeron contribution is negative, so its inclusion decreases the total result and improves agreement with data (as we can see from the plot, the solid line is closer to the data than the dashed one).
In order to correct this issue, in the resubmitted version we corrected the labels in the left panel of the Figure 1.
REFEREE:
Minor correction:
Abstract: "per cent" -> "percent"
AUTHORS:
We corrected this spelling error in the resubmitted version.

---

## Editorial Decision

published